# Fabrication and Electrolyte Characterizations of Nanofiber Framework-Based Polymer Composite Membranes with Continuous Proton Conductive Pathways

**DOI:** 10.3390/membranes11020090

**Published:** 2021-01-27

**Authors:** Takeru Wakiya, Manabu Tanaka, Hiroyoshi Kawakami

**Affiliations:** 1Department of Applied Chemistry, Tokyo Metropolitan University, 1-1 Minami Osawa, Hachioji, Tokyo 192-0397, Japan; tanaka-manabu@tmu.ac.jp; 2Research Center for Hydrogen Energy-Based Society (ReHES), Tokyo Metropolitan University, 1-1 Minami Osawa, Hachioji, Tokyo 192-0397, Japan

**Keywords:** fuel cell, polymer electrolyte membrane, polymer nanofiber, proton conductivity

## Abstract

For future fuel cell operations under high temperature and low- or non-humidified conditions, high-performance polymer electrolyte membranes possessing high proton conductivity at low relative humidity as well as suitable gas barrier property and sufficient membrane stability are strongly desired. In this study, novel nanofiber framework (NfF)-based composite membranes composed of phytic acid (Phy)-doped polybenzimidazole nanofibers (PBINf) and Nafion matrix electrolyte were fabricated through the compression process of the nanofibers. The NfF composite membrane prepared from the pressed Phy-PBINf showed higher proton conductivity and lower activation energy than the conventional NfF composite and recast-Nafion membranes, especially at low relative humidity. It is considered that the compression process increased the nanofiber contents in the composite membrane, resulting in the construction of the continuously formed effective proton conductive pathway consisting of the densely accumulated phosphoric acid and sulfonic acid groups at the interface of the nanofibers and the Nafion matrix. Since the NfF also improved the mechanical strength and gas barrier property through the compression process, the NfF composite polymer electrolyte membranes have the potential to be applied to future fuel cells operated under low- or non-humidified conditions.

## 1. Introduction

The polymer electrolyte fuel cell (PEFC) that offers electrical power generation with high efficiency and zero carbon dioxide emission is a crucial technology for resolving global energy and environmental issues [1,2,3]. For the widespread installation of PEFCs, various technical developments in PEFC components, including polymer electrolyte membranes (PEMs), are still essential [4,5]. The progress of PEMs contributes to the power generation performance and the durability and cost of the total PEFC system. For instance, PEMs that can be utilized under high temperatures (above 100 °C) and non-humidified conditions will reduce the usage and cost of the precious metal catalyst due to improved catalytic activity at high temperatures and reduce the cost and complexity of the PEFC system by eliminating the humidifiers [6,7]. However, it is hard for the conventional PEMs to attempt under such severe conditions because conventional PEMs, mostly based on sulfonated polymers, require sufficient water to dissociate protons from the acid groups and conduct protons effectively. Additionally, high gas barrier property and improved membrane stability are other issues on PEMs, especially at high temperatures.

In the past decades, an abundance of studies on PEMs, including the precise control of membrane morphology and addition of inorganic fillers, have been attempted to improve their characteristics [8,9,10,11]. Nevertheless, conventional PEMs hardly achieved high proton conductivity at low relative humidity, superior gas barrier property, and sufficient membrane stability at the same time. Much attention has recently attracted the composite PEMs containing proton conductive fillers, such as proton conductive polymer nanofibers fabricated by an electrospinning method [12,13,14,15]. Several groups including us revealed that the electrospun sulfonated polymer nanofibers showed extremely high proton conductivity because of their nano-sized structures and unique inner morphology [16,17]. Many papers have reported that the sulfonated polymer nanofibers could enhance proton conductivity as well as gas barrier property and membrane stability. Snyder and Elabd reported fabrication and fuel cell applications of Nafion^®^-based nanofibers [18]. Wang et al. attempted sulfonated poly (ether sulfone) nanofibers for through-membrane proton-conducting channels [19]. Our group also reported sulfonated polyimide nanofibers to enhance proton conductivity and electrolyte characteristics [20,21]. Most recently, Pintauro, Noto, and co-workers revealed a unique morphology and conductivity mechanism in blended polyvinylidene fluoride/Nafion electrospun nanofibers [22]. In addition to such sulfonated polymers, aromatic hydrocarbon polymers also have high potential to improve fuel cell performances [23]. For example, Jones et al. reported that electrospun polybenzimidazole (PBI) nanofiber composite electrolyte membranes showed good fuel cell characteristics due to their high membrane stability and durability [24]. Although related research has been actively studied, their proton conductivity at low relative humidity was insufficient for future PEFC operation. In our recent paper [25], we first reported the introduction of two different acid groups, sulfonic acid and phosphoric acid groups, in the nanofiber composite membranes to improve proton conductivity at low relative humidity. This strategy was based on our previous knowledge of phosphoric acid-doped blend polymer membranes composed of sulfonated polymers and PBI [26]. The phosphoric acid-doped blend polymer membranes enabled high proton conductivity at moderate relative humidity and in the absence of water under non-humidified conditions by utilizing proton hopping conduction between the two types of acid groups. By developing the idea, in our recent studies [25,27], we developed polymer electrolyte composite membranes based on acid-doped polymer nanofibers as a framework that we named “nanofiber framework (NfF)” for the proton conduction pathway. The NfF composite membranes consisted of polybenzimidazole nanofiber (PBINf) doped with phytic acid (Phy), in which six phosphoric acid groups are attached to inositol (cyclohexane-1,2,3,4,5,6-hexol) via ester bonds, and sulfonated polymer matrix (Nafion^®^ or sulfonated polyimide). The composite membrane achieved relatively high proton conductivity at low relative humidity (ca. 10^−3^ S cm^−1^ at 80 °C, 40% RH) and good fuel cell performance under low humidified conditions [25].

In this study, to improve the electrolyte characteristics of the NfF composite membranes (NfF-CMs), a compression process with high pressure (5 MPa) was attempted on the PBINfs to improve proton conductive characteristics at low relative humidity, gas barrier property, and membrane stability. The pressed PBINfs are expected to increase the effective proton conductive pathway at the interface between the nanofibers modified with phosphoric acid groups on their surface and polymer electrolyte matrix bearing sulfonic acid groups. The NfF-based composite membranes were fabricated by several procedures, including the compression processes before or after Phy-doping on the PBINfs and non-pressed conventional ones. Their mechanical strength, gas barrier property, and proton conductive characteristics, including proton conductivity under various conditions, activation energy at low humidity, and proton mobility, were investigated.

## 2. Materials and Methods

### 2.1. Materials

Phytic acid (50 wt% aqueous solution) and Nafion dispersion (Nafion^®^ DE 520, IEC = 1.03–1.12 meq g^−1^, 5 wt% in a mixture of lower aliphatic alcohols and water, contains 45% water) were purchased from TCI. Co. (Tokyo, Japan) and Sigma–Aldrich Co. (St. Louis, MO, USA) and were used as received. The electrospun polybenzimidazole nanofibers (PBINfs) were produced by Japan Vilene Company, Ltd. (Tokyo, Japan) and were used as received. A typical nanofiber diameter, thickness, and porosity of the nanofibrous membrane were 200 nm, 15 μm, and 90%, respectively.

### 2.2. Fabrication of Nanofiber Framework-Based Composite Membranes (NfF-CMs)

According to our previous study [25], a conventional Phy-PBINf/Nafion composite membrane (NfF-CM1) was fabricated. In short, the PBINfs were first immersed in a 50 wt% phytic acid (Phy) aqueous solution at 25 °C for 1 h, and then the obtained Phy-PBINf was washed with deionized water several times at 80 °C for a total 24 h to remove excess Phy. After drying in a vacuum oven at 80 °C for 12 h, an appropriate amount of the 5 wt% Nafion dispersion was carefully poured onto the Phy-PBINf in a petri dish. The solvent was then slowly evaporated under ambient conditions and the subsequent vacuum drying at 60 °C for 12 h. For the NfF-CM2, the PBINf was first pressed with 5 MPa at room temperature for 30 min by a compact pressing machine (IMC-180C, Imoto Machinery Co., Ltd., Kyoto, Japan). The pressed-PBINf was doped with Phy and composed with Nafion by the similar manners to NfF-CM1. The NfF-CM3 was fabricated in a similar way but in a different order. The PBINf was first doped with Phy, likewise the NfF-CM1, and the Phy-PBINf was pressed with 5 MPa at room temperature for 30 min. Then, the Nafion dispersion was poured into fabricating the composite membrane. The amount of Nafion dispersion was determined to fill the void in each nanofibrous membrane. After evaporating the solvent, the membranes were dried under a vacuum. The annealing process was not attempted on any membranes.

### 2.3. Characterizations

The Phy-doping level of the PBINf was gravimetrically measured from the weight change before and after the doping using Equation (1):(1)$$W\left(\%\right)=\frac{\left(W_{dop}-W_{d}\right)}{W_{d}}\times 100$$
where *W_dop_* and *W_d_* are the weights (g) of the doped and dried nanofibers, respectively. The cross-sectional scanning electron microscopy (SEM) images were taken by JXP-6100P (JEOL, Tokyo, Japan). The ion exchange capacity (IEC) value of the nanofibers and the membranes was measured using a back titration with NaCl and NaOH solutions [25,26,27]. The Phy-doping level and IEC value were estimated by the average of at least three samples or measurements. The thermal behavior of the membranes was evaluated by thermogravimetric analysis (TGA, DTG-60H, Shimadzu Co., Kyoto, Japan) from room temperature to 800 °C at a heating rate of 10 °C min^−1^ in a nitrogen atmosphere. The mechanical property was evaluated by a universal precision tester (AGS-X5kN, Shimadzu Co., Tokyo, Japan). The membranes were cut into rectangle shapes (10 × 40 mm) and set on the fixtures with a 20-mm gap. The stress-strain tests were performed at a controlled velocity of 1 mm min^−1^ until breaking under ambient conditions (27 °C and 60% RH).

For water uptake measurement, a membrane was first dried in a vacuum oven at 80 °C for 12 h and then immersed in hot water at 80 °C. After 24 h, the membrane was quickly wiped and weighed. The water uptake was calculated using Equation (2):(2)$$WU\left(\%\right)=\frac{\left(W_{s}-W_{d}\right)}{W_{d}}\times 100$$
where *W_s_* and *W_d_* are the weights (g) of the wet and dry membranes, respectively. Water uptakes at different relative humidity were also measured in a similar way using a temperature-humidity controlled chamber instead of water immersion.

The proton conductivity was measured by the electrochemical impedance spectroscopy (3532-50, Hioki Co., Tokyo, Japan) over the frequency range from 50 Hz to 50 kHz after standing the samples in a thermo-controlled humidity chamber at the appropriate temperature and relative humidity. Proton conductivity σ (S cm^−1^) of the membranes were determined from Equation (3):(3)$$\sigma =\frac{d}{t\;\cdot w}/\;R$$
where *d*, *t*, *w*, and *R* are the distance (cm) between two electrodes, thickness (cm) of the membrane, width (cm) of the membrane, and impedance value (Ω), respectively. The proton mobility $\mu_{H+}$ (cm^2^ s^−1^ V^−1^) of the membranes was estimated from Equation (4):(4)$$\mu_{H+}=\frac{\sigma \cdot V_{wv}}{F\cdot IEC\cdot W_{d}}$$
where σ, $V_{wv}$, *F,*
*IEC*, and *W_d_* are proton conductivity (S cm^−1^), volume (cm^3^) of the wet membrane, Faraday’s constant (96,500 C mol^−1^), ion exchange capacity (mmol g^−1^), and weight (g) of the dry membrane, respectively.

The gas barrier property and water vapor permeance were measured with a gas permeation measurement apparatus (RGP-3000Z, Round Science Inc., Kyoto, Japan) equipped with a gas chromatograph (GC7100, J-science Lab Co. Ltd., Kyoto, Japan). Oxygen was used as a test gas and was humidified at an appropriate relative humidity using a bubbler. The permeated oxygen and water vapor through the membrane to another side, where the helium carrier swept, was analyzed by the gas chromatography. The oxygen gas permeability coefficients $P_{O2}$ (cm^3^(STP) cm cm^−2^ s^−1^ cmHg^−1^) and the water vapor permeance $Q_{H2O}$ (g m^−2^ 24 h^−1^) were calculated by Equations (5) and (6):(5)$$P_{O2}=\frac{q}{a\cdot t}L\frac{1}{\Delta P}k\frac{273}{T+273}$$
(6)$$Q_{H2O}=\frac{q_{w}\cdot 60\cdot 60\cdot 24}{a\cdot t}k$$
where *q*, *a*, *t*, ΔP, *k*, *T*, *L*, and *q_w_* are the volume of the test gas permeated through the membrane (cm^3^), the permeation area (cm^2^), the sampling time (s), the difference in pressure by considering water vapor partial pressure (cmHg), a correction factor of the apparatus (−), the test temperature (°C), the thickness of the membrane (cm), and the weight of the water vapor permeated through the membrane (g), respectively. The measurements were performed at least three times on each sample to confirm their reproducibility.

## 3. Results and Discussion

### 3.1. Fabrication of NfF Composite Membranes

Figure 1 shows the fabrication procedures of three types of NfF composite membranes. The conventional NfF composite membrane, NfF-CM1 [25] (Figure 1a), was fabricated as follows: First, the polybenzimidazole nanofiber (PBINf) was doped with phytic acid (Phy) and washed with hot water to remove excess Phy. Then, the Nafion dispersion was poured onto the NfF (Phy-PBINf), and the solvent was slowly evaporated to provide a dense membrane. Figure 2a,b shows cross-sectional SEM images of the pristine PBINf and NfF-CM1, where the pores among the PBINfs were entirely filled with Nafion. The thickness of NfF-CM1 was ca. 30 μm, which is similar to the thickness of the PBINf. Figure 1b,c shows new procedures containing the compression processes before or after the Phy-doping to give NfF-CM2 and NfF-CM3, respectively. For the NfF-CM2, the PBINf with its thickness of ca. 30 μm was first pressed under high pressure (5 MPa). Figure 2c,d shows that the thickness of the pressed PBINf became thin to be ca. 15 μm. Subsequently, the Phy-doing and the Nafion casting were performed similarly to the NfF-CM1. The thickness of the obtained composite membrane, NfF-CM2, was ca. 18 μm (Figure 2e), which was thicker than the pristine pressed PBINf (Figure 2c). In contrast, the NfF-CM3 was obtained as a thin membrane with its thickness of ca. 14 μm (Figure 2f), which was close to that of the pressed Phy-PBINf.

### 3.2. Characterizations of the NfF Composite Membranes

The composition of each NfF-CM was evaluated by the TGA measurement (Figure 3) and weight change by the Phy-doping. As summarized in Table 1, the target (feed) ratios of NfF/Nafion in the NfF-CM1, 2, and 3 were set to be 9 wt%/91 wt% (10 vol%/90 vol%), 19 wt%/81 wt% (25/75 vol%), and 18 wt%/82 wt% (25 vol%/75 vol%), respectively. These values were determined based on the pore volume estimated from the weight and apparent volume of each NfF, the content of Phy, and densities of the materials. The actually obtained ratios of NfF/Nafion were estimated by considering the TGA results of the NfF-CMs. In Figure 3, the TGA curves showed several steps, including solvent evaporation (<300 °C), decomposition of Nafion (300–500 °C), and decomposition of the PBI-based NfF (>500 °C). The estimated ratios of NfF/Nafion in the NfF-CM1 and NfF-CM3 were almost equal to those of the target values. In contrast, the NfF-CM2 showed a relatively high NfF composition (25 wt%/75 wt%). This result indicated that the amount (volume) of Nafion was insufficient to fulfill in the pores of the NfF, suggesting the formation of voids inside the NfF-CM2. This estimation was reasonable because the appropriate amount of the Nafion dispersion poured on the NfF was set to the pore volume of the NfF by considering the thickness and porosity of the corresponding NfF and the concentration of the Nafion dispersion. In the NfF-CM2, the thickness of the pressed-PBINf was ca. 15 μm; however, the obtained NfF-CM2 showed 18 μm thickness, resulting in the amount of Nafion becoming deficient.

Table 2 summarizes the ion exchange capacity (IEC), water uptake in hot water, and mechanical properties of the NfF-CMs and the recast-Nafion membrane. The IEC values of NfF-CMs were all lower than those of the recast-Nafion membrane because of the PBI components in the composite membranes. The NfF-CM3 showed the IEC of 0.851 meq g^−1^, which was the smallest value among the membranes because the NfF-CM3 contained high NfF composition and a relatively low Phy content. The water uptake is an essential factor that influences proton conductivity and membrane stability. All the NfF-CMs showed lower water uptake than the recast-Nafion membrane, indicating that the NfFs, which were rigid frameworks, prevented the membrane’s excess swelling. The NfF-CM3 demonstrated lower water uptake than the conventional NfF-CM1 because the more considerable amount of NfF contributed to preserving membrane structure. The highest water uptake of the NfF-CM2 among the composite membranes may be derived from the unexpected pores inside that membrane. Figure 4 plots the stress-strain curves of the NfF-CMs and the recast-Nafion membrane. All the NfF-CMs showed better mechanical properties than the recast-Nafion membrane. In particular, the NfF-CM3 showed the highest maximum stress (32.6 MPa) and toughness (7.59 MJ m^−3^) (Table 2). Since mechanical strength is a crucial property in polymer electrolyte membranes for fuel cell applications, the utilization of pressed NfF is a useful approach to improve the properties.

### 3.3. Proton Conductive Characteristics of the NfF Composite Membranes

Proton conductivity is the most critical characteristic in polymer electrolyte membranes. High proton conductivity at low relative humidities (<40% RH) is incredibly desired for future fuel cell operation at high temperatures above 100 °C. Figure 5a shows the relative humidity dependence of the proton conductivity on the NfF-CMs and the recast-Nafion membrane at a constant temperature (80 °C). The NfF composite membranes, except NfF-CM2, which possessed voids inside the membrane, showed higher proton conductivity than the recast-Nafion membrane at all relative humidity ranges. Furthermore, the proton conductivity of the NfF-CM3 was higher than that of the NfF-CM1 at low relative humidities (30 and 40% RH). Further, the proton conductivity of the NfF-CM3 at 80 °C and 30% RH (4.2 × 10^−3^ S cm^−1^) was higher than the best performances in the previous reports (5.1 × 10^−4^, 1.5 × 10^−3^, and 1.7 × 10^−3^ S cm^−1^ at 80 °C and 30% RH in the references [20,21,25], respectively). Figure 5b is the temperature dependence of proton conductivity on the NfF-CMs and the recast-Nafion membrane at a constant low relative humidity (40% RH). The NfF-CM3 indicated the highest proton conductivity (ca. 10^−2^ S cm at 90 °C, 40% RH) among the membranes at all ranges of temperatures between 30 and 90 °C. Further, the activation energy of the proton conductivity on the NfF-CM3 (26.7 kJ mol^−1^) was lower than those of the NfF-CM1 (32.5 kJ mol^−1^) and the recast-Nafion membrane (45.4 kJ mol^−1^) at 40% RH. Our previous study [25] revealed that the interface between the Phy-doped PBINfs and the Nafion matrix, where phosphoric acid and sulfonic acid groups were densely accumulated, became efficient proton conductive pathways for high proton conductivity. Since the proton conduction through the nanofiber surface/interface is faster than the Nafion matrix, especially under low relative humidity conditions, the increase of nanofiber contents and better connection among the nanofibers by the compression process enhanced the proton conductivity of the composite membrane.

To understand the proton conductive characteristics of the nanofiber composite membranes, the proton mobility of each membrane was investigated as a function of its water uptake at 80 °C and various relative humidity. First, water uptakes of the membranes were measured at different relative humidities (Figure 6a). Then, the proton mobility was calculated using the proton conductivity, IEC, dry weight, and swollen volume of the membranes under each condition. As shown in Figure 6b, the NfF-CM3 showed high proton mobility even with low water uptake. These results mean that the NfF-CM3 possessed continuous and efficient proton conductive pathways compared to the conventional nanofiber composite membrane, NfF-CM1. Although the recast-Nafion membrane showed high proton mobility at high water uptake, the high water uptake was undesirable because such membranes could show low membrane stability due to their large size change in the swelling–shrinking processes. For future fuel cell operation, high proton mobility at low water uptake is favorable in the polymer electrolyte membranes. The NfF composite membranes with continuous proton conductive pathways by the compression process are possible candidates for future fuel cells operated under low or non-humidified conditions.

### 3.4. Gas and Water Vapor Permeability of the NfF Composite Membranes

The gas barrier property is also a significant property for polymer electrolyte membranes because hydrogen and oxygen gas crossover would reduce the cell potential and accelerate the degradation of membrane electrode assembly by forming reactive oxygen species, such as hydroxide radicals. Figure 7a,b shows the oxygen gas permeability and water vapor permeance of the membranes. As apparent from Figure 7a, the NfF-CM3 indicated lower oxygen permeability than not only the recast-Nafion membrane but also the NfF-CM1 and 2. In the dense NfF-CM3, the compressed NfF composed of PBI, which is known as a high gas barrier polymer, prevented the oxygen gas diffusion through the membrane. On the other hand, interestingly, the water vapor permeance through the NfF-CM3 was almost equal to other membranes. These results suggest that the water vapor could permeate through the continuous pathways, which were constructed at the interface between the nanofiber and the matrix, similar to the proton conduction. Such distinguished water permeation property is useful for utilizing generated water during the fuel cell operation, especially at low relative humidity.

## 4. Conclusions

The novel nanofiber composite membranes composed of Phy-doped PBINfs and the Nafion matrix electrolyte were fabricated through the compression process of the nanofibers. The NfF-CM3 prepared from the pressed Phy-PBINf showed higher mechanical strength and lower water uptake than the other membranes, including the conventional Phy-PBINf/Nafion, NfF-CM1, and the recast-Nafion membrane. The NfF-CM3 possessed higher proton conductivity and lower activation energy than the other membranes, especially at low relative humidity. The compression process increased the nanofiber contents in the composite membrane, resulting in the construction of the continuous and efficient proton conductive pathway at the interface of the Phy-doped PBINf and the Nafion matrix. The high proton mobility of the NfF-CM3 at a low water uptake compared to other membranes was also notable. It is considered that such outstanding characteristics can be originated from the continuously formed effective proton conductive pathway consisted of the densely accumulated phosphoric acid and sulfonic acid groups at the interface of the nanofibers and the Nafion matrix. Further study will improve the electrolyte characteristics to apply to future fuel cell operation under lower- or non-humidified conditions.

## Figures and Tables

**Figure 1 membranes-11-00090-f001:**
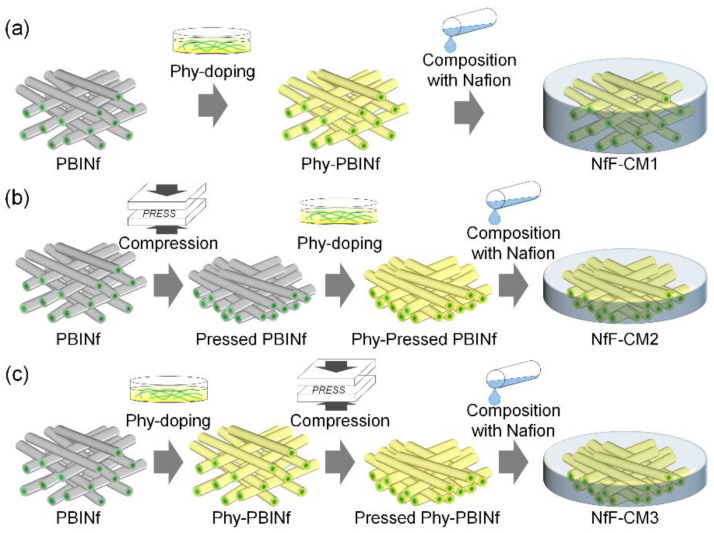
Schematic illustration of the fabrication procedures of (**a**) Nanofiber framework composite membrane 1 (NfF-CM1), (**b**) Nanofiber framework composite membrane 2 (NfF-CM2), and (**c**) Nanofiber framework composite membrane 3 (NfF-CM3).

**Figure 2 membranes-11-00090-f002:**
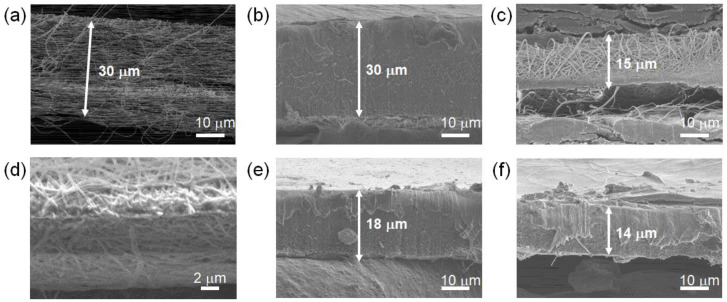
Cross-sectional SEM images of the (**a**) polybenzimidazole nanofibers (PBINf), (**b**) NfF-CM1, (**c**) pressed PBINf, (**d**) pressed PBINf (higher resolution), (**e**) NfF-CM2, and (**f**) NfF-CM3.

**Figure 3 membranes-11-00090-f003:**
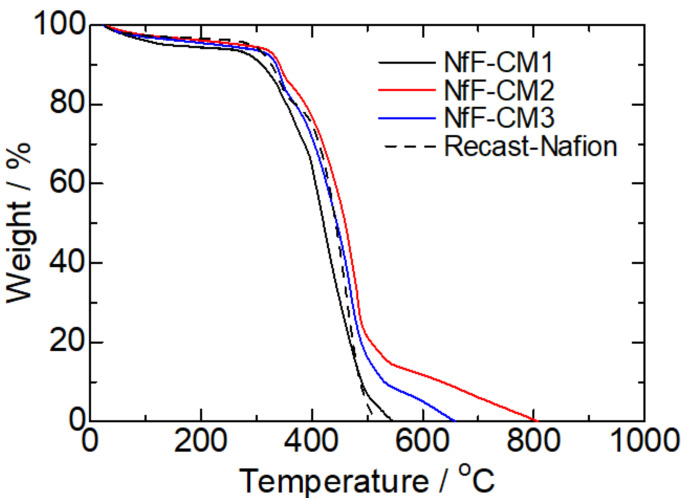
Thermogravimetric analysis (TGA) curves of the NfF-CM1, NfF-CM2, NfF-CM3, and the recast-Nafion membrane.

**Figure 4 membranes-11-00090-f004:**
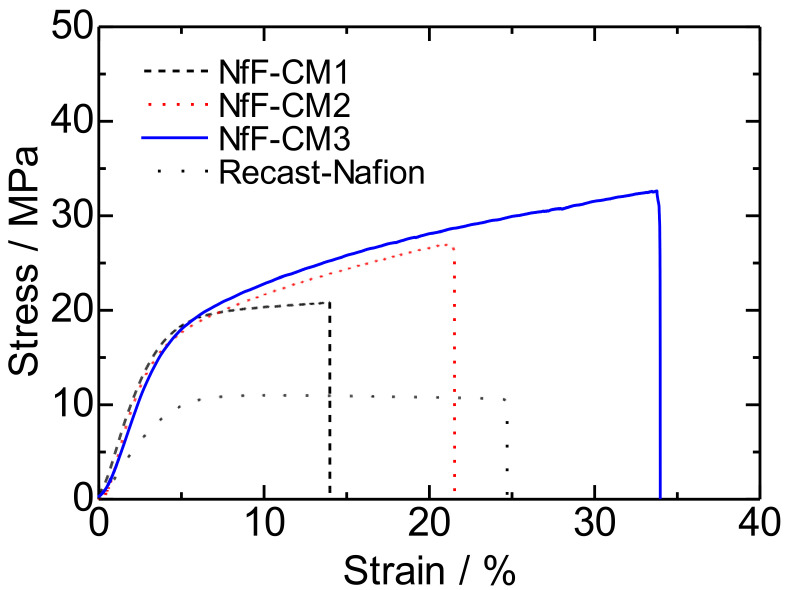
Stress-strain curves of the NfF-CM1, NfF-CM2, NfF-CM3, and the recast-Nafion membrane at 27 °C and 60% RH.

**Figure 5 membranes-11-00090-f005:**
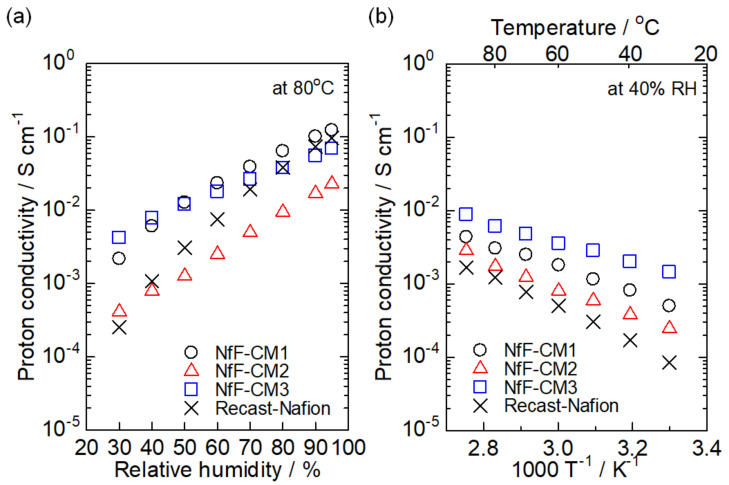
(**a**) Relative humidity and (**b**) temperature dependence of proton conductivities on the NfF-CM1, NfF-CM2, NfF-CM3, and the recast-Nafion membrane.

**Figure 6 membranes-11-00090-f006:**
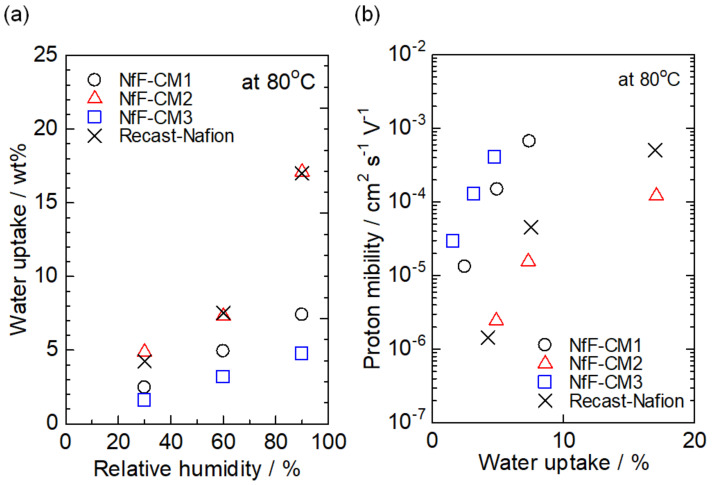
(**a**) Water uptakes and (**b**) proton mobility of the NfF-CM1, NfF-CM2, NfF-CM3, and the recast-Nafion membrane at 80 °C.

**Figure 7 membranes-11-00090-f007:**
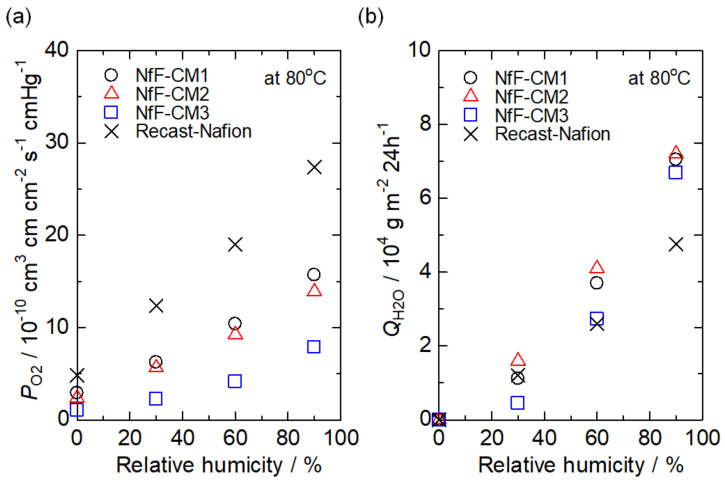
(**a**) Oxygen gas permeability and (**b**) water vapor permeance of the NfF-CM1, NfF-CM2, NfF-CM3, and the recast-Nafion membrane at various relative humidity at 80 °C.

**Table 1 membranes-11-00090-t001:** Compositions of the polymer electrolyte membranes.

Membranes	Porosity of NfF (%)	Target (Feed) Ratio of NfF/Nafion		Obtained (Estimated) Ratio of NfF/Nafion		Thickness (μm)	Phy Content (wt%) ^d^
		wt%/wt% ^a^	vol%/vol%	wt%/wt% ^b^	vol%/vol% ^c^		
NfF-CM1 ^e^	90	9/91	10/90	9/91	11/89	30	2.0
NfF-CM2 ^f^	75	19/81	25/75	25/75	25/58 + void	18	5.0
NfF-CM3 ^g^	75	18/82	25/75	19/81	24/76	14	3.0
Recast-Nafion	−	0/100	0/100	0/100	0/100	42	−

^a^ Calculated from the target ratio of nanofiber framework (NfF)/Nafion (vol%/vol%) and densities of polymers; ^b^ determined by the thermogravimetric analysis (TGA) measurement; ^c^ estimated based on the obtained ratio of NfF/Nafion (wt%/wt%); ^d^ weight ratio of phytic acid in the membranes calculated from the weight ratio of NfF and weight change from polybenzimidazole nanofibers (PBINF) to phytic acid-doped polybenzimidazole nanofibers (Phy-PBINF); ^e^ nanofiber framework composite membrane 1 (NfF-CM1); ^f^ nanofiber framework composite membrane 2 (NfF-CM2); ^g^ nanofiber framework composite membrane 3 (NfF-CM3).

**Table 2 membranes-11-00090-t002:** Ion exchange capacity (IEC), water uptake, and mechanical properties of the polymer electrolyte membranes.

Membranes	Expected IEC (meq g^−1^) ^a^	IEC (meq g^−1^) ^b^	Water Uptake (%) ^c^	Mechanical Properties ^d^		
				Maximum Stress (MPa)	Young modulus (MPa)	Toughness (MJ m^−3^)
NfF-CM1	0.97–1.05	0.984	18 ± 2	19.6	519	2.20
NfF-CM2	0.91–0.98	1.05	22 ± 5	26.7	564	4.09
NfF-CM3	0.88–0.95	0.851	14 ± 2	32.6	460	7.59
Recast-Nafion	1.03–1.12	1.089	34 ± 5	11.0	233	2.24

^a^ Estimated by weight fraction of Nafion and free acid groups of phytic acid on PBINf; ^b^ determined by a titration method; ^c^ weight change after immersion in water at 80 °C for 24 h; ^d^ at 27 °C and 60% RH.

## Data Availability

Not applicable.

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
