# Peer review of "Fabrication and Electrolyte Characterizations of Nanofiber Framework-Based Polymer Composite Membranes with Continuous Proton Conductive Pathways"

_membranes, 2021, doi:10.3390/membranes11020090_

Round 1

Reviewer 1 Report

The manuscript describes the preparation of composite membranes consisting of phytic acid doped PBI nanofiber mats and Nafion. In contrast to previous work, the effect of compaction on the nfm was investigated.

It would have been nice if the paper could have included fuel cell results.
Besides this, the paper is well written and thoroughly investigated.
Nevertheless, I have some comments which should be considered before publication.

  1. Line 66: please elaborate, that the 6 phosphoric acid groups are attached to inositol (cyclohexane-1,2,3,4,5,6-hexol) via ester bonds.
  2. Please mention also the work by Deborah Jones in the introduction, she showed at some conferences the preparation of PBI/Nafion composite membranes, in which Nafion and PBI show intimate interaction by ionic bonds. Some data can be found in the patent WO2016/020668 “Membrane”, https://patentscope.wipo.int/search/en/detail.jsf?docId=WO2016020668&tab=PCTDESCRIPTION
  3. The patent shows also conductivity of 173 mS/cm at 80C, 95%rh, which seems to be much higher than for your membranes. What could be the differences?
  4. Please state if you used Nafion EW1000 or NAfion EW1100.
  5. PBI nanofiber mats are usually prepared by electrospinning form DMAc solution. Although the fibers have a large surface, it could be that DMAc remains inside. For membranes, this DMAc amount can be as high as 50-60 wt%. When you measured the dry weight (equation 1, phytic acid uptake), did you wash the membrane with 80C water for 24 hours or with acetone over night or another similar process to remove residual DMAc, or did you check by nmr the contents of DMAc? Because DMAc may leach out during doping and then would lead to too low uptake values.
  6. Please correct eq. 3 (/R, not xR)
  7. Please inform the porosity of the nanofiber mats before pore filling, for example in Fig 1. Or in Table 1.
  8. It would be nice to see also a SEM image of pristine PBI fiber mat, and perhaps a higher resolution picture of Fig 2b (fiber mat after compaction).
  9. Line 186: It would be better to write steps, not decomposition steps.
  10. Please explain more in detail how you prepared the membranes. Did you wipe the excess solution away? Because when you drop Nafion dispersion on the fiber mat and evaporate the solvent, the ratio of PBI/Nafion cannot change from feed to membrane, as discussed. A different ratio would only be seen if you wipe solution away. If you did not do this, there should be a Nafion surface layer on top of the reinforced layer, but ratio is same as the feed ratio.
  11. line 198: Table 2, not 3
  12. Table 1, first line: PBI (density 1.3 g/ml) / Nafion (density 1.92 g/ml) combined in a weight ratio of 9/91 should give a volume ratio of 13/87, but you state 10/90. Please correct this.
  13. Please add a column for expected IEC, taking into account the weight fraction of Nafion and amount of NH groups of PBI which deprotonate phytic acid and phytic acid.
  14. Please add standard deviation values for values in Table 2.
  15. The elongation at break value for recast Nafion is very low. The reason is that you did not anneal the membranes. To get good Nafion membranes, they should be treated e.g. at 120C for 1 hour. Why did you not anneal the membranes? Please comment on this in the paper.
  16. In Fig 4, the 3 straight curves cannot be distinguished in paper form, it would be nice if you could use different dotted lines or thicknesses.
  17. Why is conductivity different for 80C, 40%rh in Fig 5a and Fig 5b?

Author Response

We very much appreciate your constructive comments and valuable indications and questions on our manuscript. We have extensively revised the manuscript in response to your questions and comments.

Reviewer 2 Report

In this work, nanofiber composite membranes composed of Phy-doped PBINfs and the Nafion matrix electrolyte were fabricated, and the results are of interest for the readers. However, the discussions on the results could be more well organized. In the discussion part, a more interesting discussion should be explored, especially comparing with the literature data (The results are better or worse than the reported data?). Moreover, the introduction part could include a small interesting review of similar composites.

Author Response

We very much appreciate your review. Your positive and constructive comment on our manuscript. We have revised the manuscript in response to your comments.

Round 2

Reviewer 2 Report

The manuscript can be accepted in presented form